# Over-the-counter carrageenan-based sprays may interfere with PCR testing of nasopharyngeal swabs to detect SARS-CoV-2

Taylor Corocher[1,2*], Kira Edwards[1,2], Yvonne Hersusianto[1,3], Donald Campbell[4,5], Hui Yin Lim[1,2,6,7,8], Paul Monagle[9,10,11,12], Prahlad Ho[1,2,6,7,8]

1 Northern Clinical Diagnostics and Thrombovascular Research (NECTAR) Centre, Northern Health, Melbourne, Victoria, Australia, 2 Northern Pathology Victoria, Northern Health, Melbourne, Victoria, Australia, 3 Infectious Diseases, Northern Health, Melbourne, Victoria, Australia, 4 Hospital without Walls, Northern Health, Melbourne, Victoria, Australia, 5 Department of Medicine, Southern Clinical School, Monash University, Clayton, Victoria, Australia, 6 Department of Haematology, Northern Health, Melbourne, Victoria, Australia, 7 Department of Medicine, Northern Health, University of Melbourne, Melbourne, Victoria, Australia, 8 Australian Centre for Blood Diseases, Monash University, Clayton, Victoria, Australia, 9 Department of Paediatrics, University of Melbourne, Melbourne, Victoria, Australia, 10 Murdoch Children's Research Institute, Melbourne, Victoria, Australia, 11 Department of Haematology, Royal Children's Hospital, Parkville Victoria, Australia, 12 Kids Cancer Centre, Sydney Children's Hospital, Randwick, New South Wales, Australia

* taylor.corocher@nh.org.au

## Abstract

Carrageenan-containing nasal sprays, available over-the-counter (OTC), are often marketed as having anti-viral effects. Carrageenan belongs to the glycosaminoglycan family alongside heparin, and heparin is known to inhibit real-time quantitative polymerase chain reaction (RT-qPCR) in nasopharyngeal swabs used to detect SARS-CoV-2. As heparin and carrageenan share structural similarities, this work aimed to investigate the interferent effect of carrageenan on RT-qPCR for SARS-CoV-2 detection across 4 different diagnostic platforms. This work demonstrated that in the presence of carrageenan samples return inaccurate and invalid results on the Seegene STARlet, while qualitative accuracy was maintained on the Cepheid GeneXpert, Roche Cobas LIAT, and Hologic Panther Aptima. Evidence of carrageenan interference on SARS-CoV-2 testing was consistent across two OTC brands and research-grade reconstituted iota-carrageenan, with 80% of results returning invalid regardless of the carrageenan formulation added to the samples. Further, a preliminary *in vivo* interference study demonstrated an increased Ct value within 15 minutes of carrageenan dosage, with Ct values restored 60 minutes post-application. A direct comparison of carrageenan- and heparin-mediated PCR interference demonstrated that carrageenan PCR interference occurs to a lesser degree, but is not reversible by the addition of heparinase I. As carrageenan is available OTC, interference with PCR testing that causes an increase in false negative results could lead to accidental spread of disease and could therefore have significant public health impacts on community testing of respiratory infectious diseases via PCR.

**Data availability statement:** All relevant data for this study are publicly available from the figshare repository (https://doi.org/10.6084/m9.figshare.28004588.v1).

**Funding:** Initials of the authors who received each award Paul Monagle, Donald Campbell Grant numbers awarded to each author N/A The full name of each funder The Victorian Government Department of Jobs, Skills, Industry and Regions (DJSIR) URL of each funder website Did the sponsors or funders play any role in the study design, data collection and analysis, decision to publish, or preparation of the manuscript? No.

**Competing interests:** The authors have declared that no competing interests exist.

**Abbreviations:** CG,Carrageenan; OTC,Over-The-Counter; GAG,Glycosaminoglycan; RT-qPCR, Real-Time Quantitative Polymerase Chain Reaction; NP, Nasopharyngeal; CT, Cycle Threshold; HSV, Herpes Simplex Virus; HPV, Human Papillomavirus; VZV, Varicella-Zoster Virus; UTM, Universal Transport Medium;IC, Internal Control.

## Introduction

One of the critical components in the successful management of the SARS-CoV-2 pandemic is the effective and accurate PCR-testing for viruses. Our group is investigating use of a heparin-containing nasal spray as a prophylactic against SARS-CoV-2 in the IntraNasal Heparin Trial (INHERIT) (NCT05204550). Our recent work demonstrated that heparin causes interference in SARS-CoV-2 PCR assays, resulting in an increase in invalid and false negative results when testing nasopharyngeal (NP) swabs immersed in heparin, interference which can be prevented through treatment with the enzyme heparinase I [1].

Heparin is structurally similar to the algae-derived glycosaminoglycan (GAG) carrageenan (CG) [2], which is available in multiple markets as an over the counter (OTC) nasal spray, purporting to have antiviral effects. The global CG market was valued at USD$871 million in 2022 with a predicted growth rate of 5.4% by 2030 [3]. The most commonly available types of CG are, iota-, kappa- and lambda-CG. Studies of iota-CG have demonstrated antiviral activity against herpes simplex virus (HSV), human papillomavirus (HPV), varicella-zoster virus (VZV), and human rhinoviruses *in vivo* and *in vitro* [2,4–6]. Iota-CG may act to prevent viral binding to heparan sulfate proteoglycans in host tissues, inhibiting viral replication [7,8]. Whether CG interferes with polymerase chain reaction (PCR) is unknown, but the structural similarity to heparin suggests it may have a similar effect. Therefore, evidence of CG-mediated PCR interference in NP samples could potentially have substantial public health implications, as false negative results could lead to an increase in the accidental spread of respiratory illnesses such as SARS-CoV-2, which is especially significant given that CG-containing nasal sprays are readily available, and marketed for respiratory symptoms. This study aims to evaluate CG-mediated PCR interference on several SARS-CoV-2 diagnostic testing platforms.

## Methods

### Diagnostic tools, patient samples and ethical approval

Northern Pathology Victoria (NPV), Australia, uses several platforms for SARS-CoV-2 detection: the Seegene (In Vitro Diagnostics, Korea) STARlet liquid handling workstation in combination with the Seegene Allplex SARS-CoV-2 assay (amplification of the SARS-CoV-2 E-gene, N-gene and RdRP/S-gene), plus either the Seegene STARMag extraction kit, or the TANbead nucleic acid extraction kit in conjunction with the MAELSTROM 9610 nucleic extractor (both Taiwan Advanced Nanotech inc, Taiwan), with PCR amplification performed on the C1000 touch thermocycler (Bio-rad, USA). Other platforms used include the Panther system with the Aptima SARS-CoV-2 Assay (Hologic, USA), the GeneXpert Xpress CoV-2 cartridge (Cepheid, USA), and the Cobas LIAT with the Cobas SARS-CoV-2 and Influenza A/B Assay (Roche Diagnostics, Switzerland).

Residual samples collected from SARS-CoV-2 positive patients were retained at NPV after routine clinical testing; these samples were initially collected by NP swab sampling into 3 mL Copan Universal Transport Medium (UTM; Copan Diagnostics, Italy), and stored at -80°C after initial testing. A waiver of consent for the use of residual samples for this research was granted by the Alfred Hospital Human Research Ethics Committee (Project Number: 67614). Samples used in the following experiments were accessed between 06/02/2023 and 21/07/2023.

### Carrageenan interference across diagnostic testing platforms

To investigate if CG-mediated PCR interference occurred, a pool of residual samples from SARS-CoV-2 positive patients (with an aggregate Ct value of 25) was prepared and separated into 3 mL aliquots. The swab submersion method was performed as described in [1], using FLO Travel nasal spray (1.6 mg/mL CG, consisting of 1.2 mg/mL iota-CG and 0.4 mg/mL

kappa-CG, Aspen Pharmacare, Australia; unless stated otherwise). Each sample was tested using the assays described, with a dilution gradient of FLO Travel nasal spray.

## Carrageenan interference consistency across different formulations

In a second experiment, consistency of CG interference was investigated using two brands of commercially available nasal sprays: (FLO Travel nasal spray 1.6 mg/mL and BETADINE Cold Defence nasal spray 1.2 mg/mL iota-CG, iNova Pharmaceuticals, Singapore), and reconstituted iota-CG powder (Sigma-Aldrich, USA) in 0.9% NaCl (to 1.6 mg/mL). Ten SARS-CoV-2 positive samples were pooled (with an aggregate Ct value of ~ 25 across genes tested) and then diluted to a total volume of 10 mL in UTM and split into three 3 mL aliquots, with the remaining 1 mL set aside as the sample-only control. For each 3 mL aliquot, the swab submersion method was performed with a 1/8 dilution of each CG formulation and tested on the Seegene platform with the Allplex SARS-CoV-2 assay. The Allplex SARS-CoV-2 assay returns Ct values for three genes specific to SARS-CoV-2 (E-gene, N-gene, RdRP/S-gene), the variation in the Ct value of the sample-only was compared to the sample after being spiked with carrageenan.

As the different brands of nasal sprays tested had different CG concentrations, to enable equal comparison of the degree of interference between formulations, a series of dilutions was used. For clarity, the dilution factor used will be referred to, rather than the heparin or CG concentrations.

## A direct comparison of heparin and carrageenan PCR interference and the efficacy of heparinase I against carrageenan

A comparison of heparin (5,000 IU/mL; Sigma Aldrich, USA) and carrageenan (FLO Travel nasal spray 1.6 mg/mL) was performed by testing a dilution series of each interferent on a pool of residual SARS-CoV-2 positive material. Nucleic acid was extracted on the Seegene STARlet using the STARMag extraction kit, and extracted samples were then treated with 10 uL of 0.4 IU/mL heparinase I (IBEX technologies, Canada) (incubated at room temperature for 30 minutes), and PCR setup was performed using the Seegene Allplex SARS-CoV-2 assay on the C1000 touch thermalcycler. Samples were tested in triplicate and run in parallel to avoid any confounding variables. To ensure the viability of heparinase I used, control samples were prepared from pooled residual SARS-CoV-2 material and included in each assay run. These controls contained 1) sample-only; 2) sample + spiked heparin; 3) sample + spiked heparin + heparinase I treatment. Control 1 acts as the baseline for an inhibited reaction, control 2 demonstrates the increased Ct value due to the addition of heparin, and control 3 is expected to return to within 2 Ct values of control 1 if heparinase I is active.

To test the efficacy of heparinase I (IBEX technologies, Canada) against iota-CG, the swab submersion method was performed as described in [1], using FLO Travel nasal spray. Nucleic acid extraction was performed using the STARMag extraction kit on the Seegene STARlet. Subsequently, 10 uL of 0.4 IU/mL heparinase I was added to extracted nucleic acid and incubated at room temperature for 30 minutes, before PCR set up on the Seegene STARlet using the Allplex SARS-CoV-2 assay on the C1000 touch thermal cycler. Heparinase I was reconstituted in a buffer made of 20 mM TRISMA base, 600 mM NaCl and 150 mM $CaCl_2$ adjusted to a pH of 7.0 with HCl.

## *In vivo* interference testing

To determine the effects of CG used on SARS-CoV-2 testing *in vivo*, NP swabs were collected from an author who was using FLO Travel nasal spray (KE). A baseline swab was collected prior to administration of the nasal spray, swabs were then collected at 15 minutes, 30 minutes, 45 minutes, 60 minutes, 2 hours, 4 hours, 8 hours, 16 hours, and 24 hours post-dose; this

was repeated on three separate occasions (biological replicate, n = 3). Each swab was collected in 2 mL of COPAN UTM (1 mL removed prior to collection) and later spiked with 1 mL of pooled SARS-CoV-2 positive material. Different SARS-CoV-2 positive pools were used to spike each set of biological replicates, samples were stored at -80°C and defrosted at RT prior to analysis in technical triplicate using the STARMag extraction and Allplex SARS-CoV-2 assay kits, as per manufacturers' instructions.

### Statistical analysis

Ct values presented in this work are an average of three technical triplicates. For the *in vivo* interference testing, the change in Ct value was normalized by subtracting the baseline sample Ct values from the Ct values of each time point post-dose samples. Confidence intervals (CI) were calculated using descriptive statistics in Prism (GraphPad, Dotmatics). On the Seegene platform, Ct values > 40 cycles are considered invalid, which is defined as a failure of the internal control (IC) to detect amplification of a known sequence at a predetermined concentration.

## Results

### Analysis of carrageenan interference across diagnostic testing platforms

Pooled SARS-CoV-2 positive material was divided into aliquots and spiked with varying dilutions of FLO Travel nasal spray, ranging from the stock concentration (1.6 mg/mL) to a 1/32 dilution. CG spiked samples were then tested across different diagnostic platforms for any inhibitory effect as described in the methods.

Where the CG concentration was greater than ¼ dilution, analysis with both the STAR-Mag and Tanbead extraction kits returned invalid results (Fig 1). In contrast, the Hologic Panther, GeneXpert, and LIAT platforms correctly detected SARS-CoV-2 genetic material at all concentrations of CG tested (Fig 1).

However, correctly determining the presence or absence of SARS-CoV-2 in a sample does not confirm the absence of CG-mediated PCR interference. Analysis of the Ct values from the GeneXpert (Fig 2A) and Seegene STARMag assays (Fig 2B) demonstrate an increasing delay in Ct values with increasing CG concentration, with Ct values increasing by ten when samples are spiked with undiluted CG solution (in comparison to the sample-only control). No evidence of interference was seen at CG concentrations lower than the 1/4 dilution. However, the Seegene STARMag appears to be more susceptible to the effects of PCR interference at lower concentrations of CG with invalid results in the presence of undiluted CG.

As carrageenan nasal sprays consist of a mixture of different carrageenan isotopes (mainly iota and kappa) the consistency of CG-mediated interference was investigated across three available brands: FLO Travel nasal spray, Betadine Cold Defence spray, and Sigma CG reconstituted in 0.9% NaCl (Table 1). The residual material from 10 individual patient samples was tested without carrageenan, and separated into 3mL aliquots that were spiked with each formulation individually as described in the methods.

The SARS-CoV-2 positive agreement for each formulation of CG compared to the 'Sample-only' control are presented in Table 1, showing a consistent 20% positive agreement across tested formulations.

### Direct comparison of heparin and carrageenan interference on SARS-CoV-2 testing, and subsequent heparinase I treatment

Finally, a comparison of the degree of heparin and CG interference was performed using pooled SARS-CoV-2 positive material (as per the methods). Tested concentrations of CG and heparin were the stock concentration, 1/8, and 1/32 dilutions (Fig 3). Both heparin and

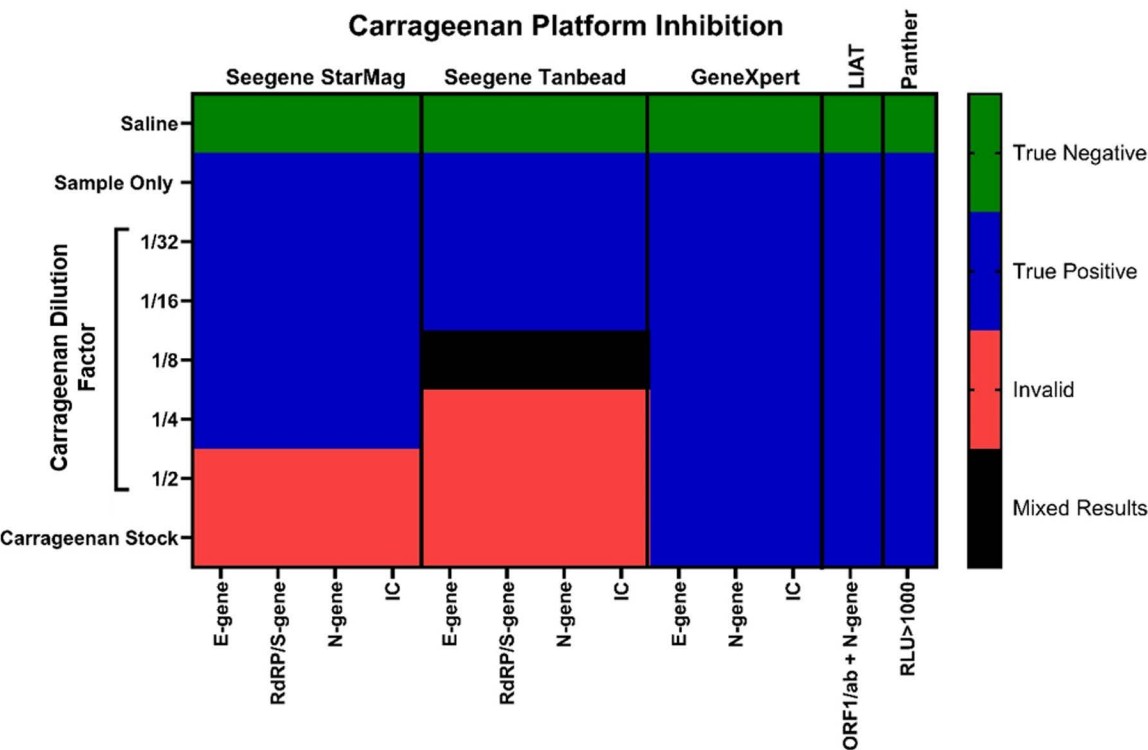

**Fig 1. The effect of carrageenan on different diagnostic platforms.** A pool of SARS-CoV-2 residual patient material was divided into 3 mL aliquots. The swab submersion method was performed using a stock concentration of carrageenan, a 1/2 dilution, 1/4 dilution, 1/8 dilution, 1/16 dilution, and a 1/32 dilution. Samples were tested on the GeneXpert, the Roche Liat, the Hologic Panther and the Seegene STARlet in combination with the STARMag or TanBead extraction kits (as indicated). Mixed results were defined as differences in results across the technical triplicates (e.g. negative/positive/one-gene positive etc).

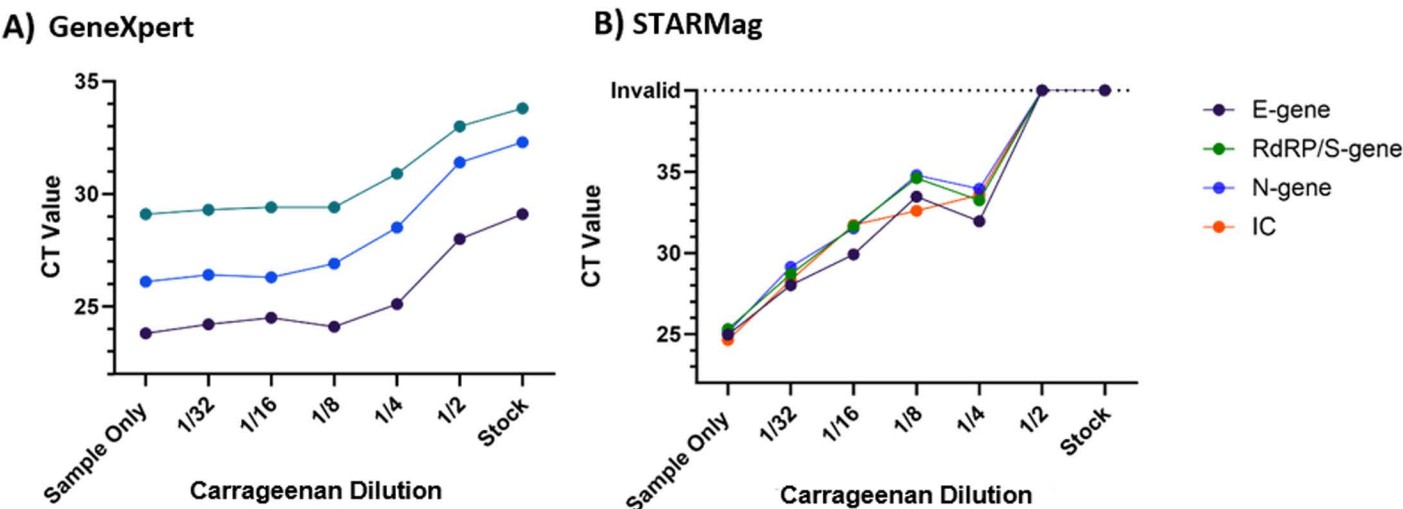

**Fig 2. The effect of CG interference on Ct values.** A pool of SARS-CoV-2 was created from residual patient samples and divided into 3 mL aliquots. CG was diluted from stock (1.6 mg/mL) to 1/32 of the recommended dosage concentration. The swab submersion method was performed and samples were tested on A) Cepheid GeneXpert with the Xpress CoV-2/Flu/RSV cartridge, and B) using the STARMag extraction kit and Allplex SARS-CoV-2 assay.

**Table 1. SARS-CoV-2 positive agreement of carrageenan brands (1/8 dilution), performed using the STARMag extraction kit and the Allplex SARS-CoV-2 assay.**

|  | Sample-only Control | Sigma Carrageenan | FLO-travel | Betadine |
|---|---|---|---|---|
| Positive | 10 | 2 | 2 | 2 |
| Invalid | 0 | 8 | 8 | 8 |
| Positive agreement % |  | 20% | 20% | 20% |

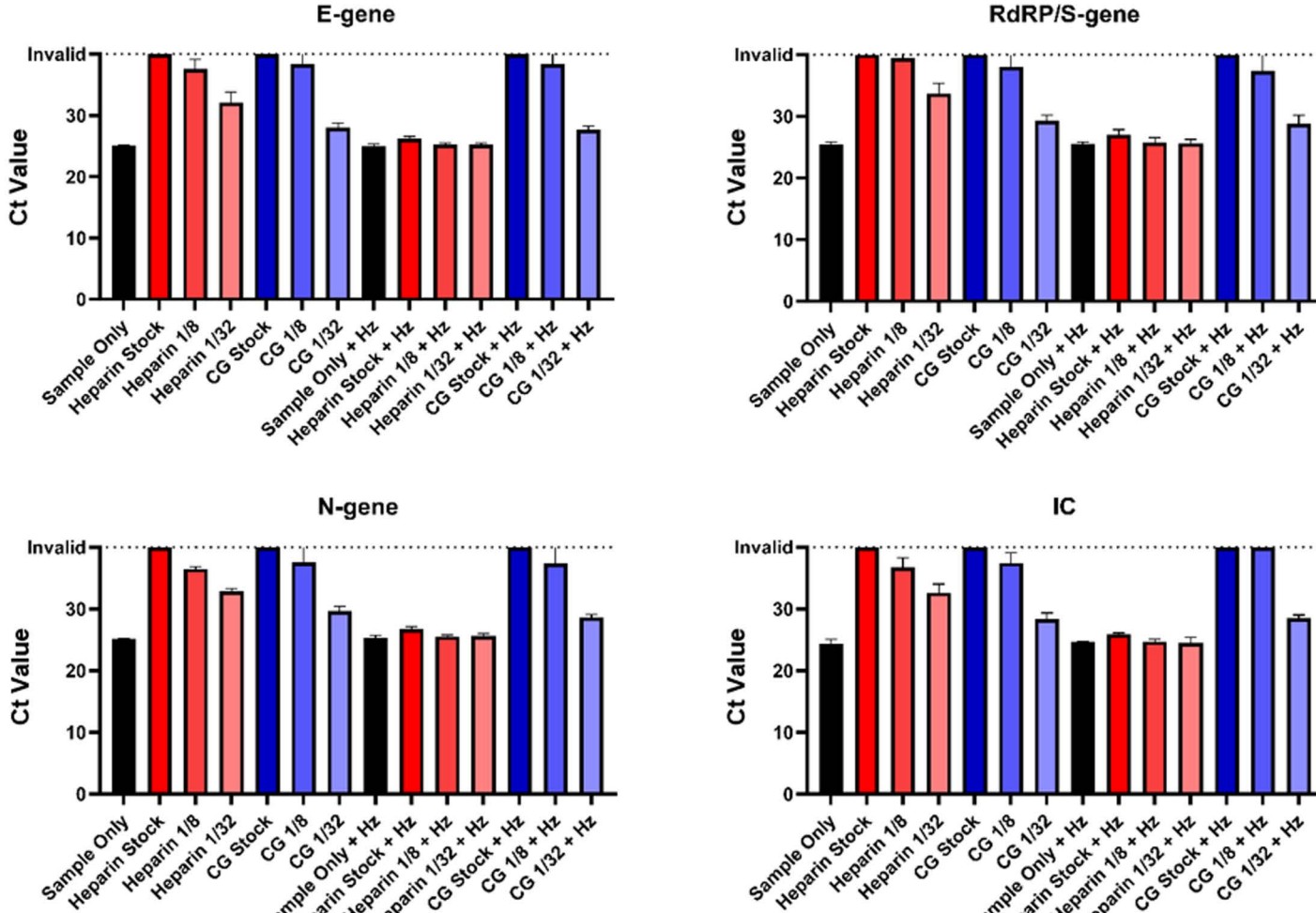

**Fig 3. A comparison of PCR interference between heparin and carrageenan and the effect of heparinase I treatment.** A pool of residual SARS-CoV-2 material was split into 3 mL aliquots. For each aliquot, the swab submersion method was used with either heparin (red bars) or CG (blue bars) at the following concentrations: stock (1.6 mg/mL), 1/8 dilution, 1/32 dilution. For each dilution, 6 aliquots were prepared alongside a sample-only control. Heparinase I was added to half the aliquots prepared after nucleic acid extraction (using the STARMag extraction kit) and incubated at room temperature for 30 minutes. PCR setup was performed with the Allplex SARS-CoV-2 assay and run on the thermocycler. (CG = Carrageenan, Hz = Heparinase I, IC = Internal Control).

CG, at stock and 1/8 dilutions, returned either an invalid result or a positive result with Ct values approaching the cut-off threshold of 40 – a marked increase when compared to the sample-only control Ct values of ~ 25 Ct. The 1/32 dilution of CG had an internal control (IC) Ct of 28.36, vs the sample-only Ct of 24.34. In contrast, the 1/32 dilution of heparin had an IC

Ct of 32.61 (S4 Table). The trend of higher Ct values in the presence of heparin compared to CG was similar across all genes tested (S1–3 Tables).

After treatment with heparinase I, samples containing heparin resulted in Ct values comparable to the sample-only control, while treated samples containing CG remained at higher Ct values.

### *In vivo* interference testing

A pilot experiment was performed as proof-of-concept to determine if the interfering effect of CG would be seen *in vivo*, and to examine how long after administration of a single dose of a CG-containing nasal spray this interference was evident in resulting Ct values. To assess changes in Ct value over time, post-dose sample Ct values were normalized by subtraction of the baseline (0) time point Ct value (Fig 4). There was an increase of ~ 3-4 Ct values across all three genes tested and the IC using the Allplex SARS-CoV-2 assay on samples collected 15 minutes post-dose. This increase in Ct values was no longer evident at the 30 minutes post-dose time-point.

Additional evidence of *in vivo* interference comes from our clinical trial (INHERIT) examining the use of intranasal heparin as a prophylactic to prevent transmission of SARS-CoV-2 within a household. Having defined the degree of heparin-mediated PCR interference prior to commencement of the INHERIT study [1], strict protocols were developed for the processing and analysis of samples collected from trial participants to ensure the accuracy of trial data in the presence of a known PCR-interferent. Because our previous investigations showed that in heparin-spiked samples Ct values post-heparinase I treatment are within 1-3 Ct values of their initial Ct value when un-spiked, we set a threshold IC Ct value of < 27 for trial runs (regardless of whether or not SARS-CoV-2 is detected) as an indication of potential interferent presence, and, along with control samples, as confirmation that the heparinase I treatment has worked. Samples with ICs outside this range undergo repeat testing to satisfy this requirement.

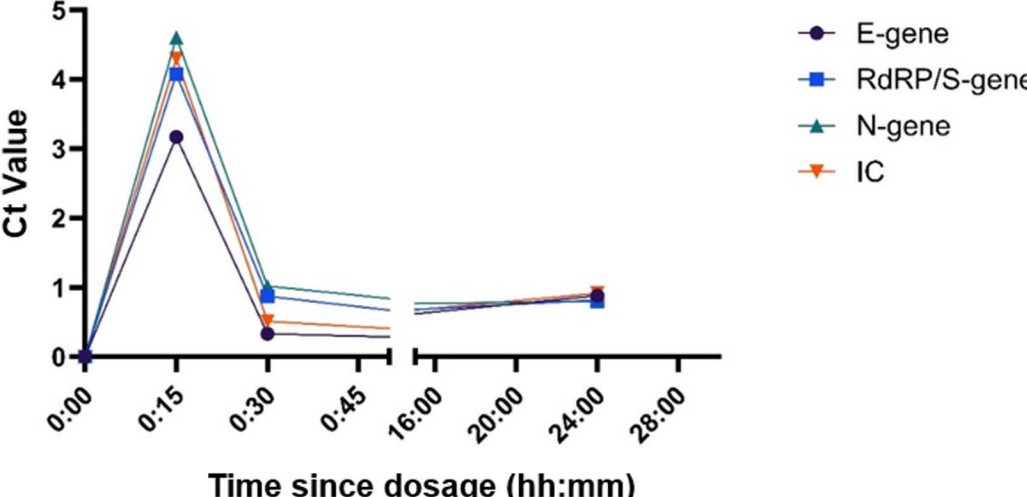

**Fig 4. Determination of the presence of *in vivo* CG-mediated PCR inhibition.** A baseline swab was collected (time 0), followed by administration of FLO Travel nasal spray. Subsequent swabs were collected at 15 minutes, 30 minutes, 45 minutes, 60 minutes, 2 hours, 4 hours, 8 hours, 16 hours, and 24 hours post-dose. Each swab was collected in 2 mL of UTM and spiked with 1 mL from pooled SARS-CoV-2 positive material. Results shown are the mean of 3 biological replicates, normalized against the sample-only control, all samples were run in three technical replicates. Data points represent the average of three biological replicates (n = 3). (IC = Internal Control).

To date, the majority of participant samples analyzed pass this IC threshold test, with those that initially fail passing upon repeat testing. However, our attention was drawn to a subset of samples from two participants, from independent households, which consistently returned IC Ct values > 27, despite repeat testing, and with no other samples in those runs showing evidence of systemic PCR interference.

Upon investigation, it was discovered that both participants had been using a nasal spray containing CG within 24 hours prior to consenting to trial participation, collection of the screening NP swabs by the study nurse, and prior to starting treatment with the study drug. Results for both participants showed IC Ct values > 27 at screening, and in swabs collected on subsequent days after starting treatment the degree of interference as seen in the Ct values decreased (S5 and S6 Tables).

Two swab types are used for sample collection during the trial, the traditionally used FLOQSwab (Copan Diagnostics, Italy) and a two-pronged Rhinoswab (Rhinomed, Australia). Generally, higher Ct values were seen with Rhinoswab-collected samples. The difference in degree of interference seen between sample collection methods could be due to the extended time that the Rhinoswab has contact with the nasal mucosa (~1 minute) when sampling, compared to the FLOQSwab (~10 seconds per nostril).

## Discussion

### The effect of carrageenan across SARS-CoV-2 diagnostic testing platforms

The data presented in Fig 1 demonstrates that CG-based nasal sprays, which are readily available OTC and marketed as preventing or lessening the symptoms of respiratory illness, interfere with PCR testing on a number of commercially available diagnostic testing platforms.

Qualitatively, we demonstrate that the presence of CG has an interfering effect (Fig 1) in samples tested, particularly affecting the STARMag extraction and Allplex SARS-CoV-2 assay kits. Similarly, despite not showing qualitative discrepancies in results, there was a significant increase in Ct values in the assays up to 10 cycles. A potential reason qualitative results were preserved may be the low starting Ct value of our test SARS-CoV-2 samples (25 compared to 32 in our previous studies). Hence, the clinical impact of this interference is significant and could significantly impact results in patients, with a greater degree of impact seen in samples with higher starting Ct values, which is an indication of a lower viral titer in the sample.

We have also noted a differential impact of the different assays. This may be in part due to the differences in the underlying technology used. Panther, for example, appears to be minimally impacted by both CG and heparin interference. One of the reasons hypothesized is the specificity of the Panther's magnetic beads which contain oligonucleotides complementary to SARS-CoV-2 nucleic acid, instead of relying only on the electromagnetic attraction of nucleic acid to the beads.

Similarly, when looking at the Ct values returned by the GeneXpert assay (Fig 2A), we can see an increase when CG is present, indicating interference, however the degree of interference was not sufficient to cause an invalid or false negative result. However, the comparative increase in Ct values between the GeneXpert and Seegene Allplex SARS-CoV-2 assays was much smaller (Fig 2B), indicating that the Allplex assay may be more susceptible to interfering agents than the GeneXpert. This is consistent with our data from the heparin study [1]. Unfortunately, due to the proprietary nature of diagnostic assays and platforms, we were unable to determine if they had different technical characteristics that may have led to these differences in results.

Furthermore, we also confirmed that this inhibitory PCR effect is seen across all different subtypes of CG, with all three formulations of CG at 1/8 dilution failed to detect SARS-CoV-2 in 8 out of 10 samples. This data suggest that it is a class effect of CG, independent of any compounds used, and that regardless of the brand used diagnostic accuracy could be reduced if patients are actively using a CG-based nasal spray.

The difference in Ct values described above, while small, may both have a significant impact in monitoring public health measures and also affect results from clinical studies. There are a number of studies in the literature investigating CG as potential antiviral treatment, where viral status is determined using PCR methodologies, and thus, used to assess the efficacy of CG as an antiviral [9–14]. As our data shows that CG has an interfering effect on PCR that artificially elevates Ct values, this suggests the results of these studies should be interpreted with a degree of caution, as findings of higher rates of negative test results or higher Ct values post-treatment could be due to the impact of CG interference on PCR efficiency, and not necessarily an indication of treatment efficacy. Likewise, studies that do not account for differences in PCR efficiency, such as by presenting the IC Ct values, or where results are not normalized against an endogenous control of known concentration, should also be interpreted with caution.

## A comparison of heparin and carrageenan PCR interference

When directly comparing CG and heparin interference on the same sample material, CG had a lower interference impact, and its interferent effect was only noted at higher concentrations (Fig 3). Part of the reason for this may be the minor structural differences seen between heparin and carrageenan. For example, heparin is a highly negatively charged biomolecule, with a density charge of 3.3 [15], so it is likely to out-compete nucleic acids when binding to magnetic beads, however iota-CG has a density charge of only 1.53 [16], which suggests it would be less successful in displacement of nucleic acids when binding to magnetic beads, explaining the lower degree of interference seen in comparison to heparin.

Given the structural similarities with heparin, we initially hypothesized that heparinase may be able to reverse the interference. Unfortunately, heparinase I treatment of CG-containing samples showed little change in Ct values compared to the untreated CG-containing samples (Fig 3), thus ruling it out as a method of removing interference. It is difficult to comment on the statistical significance of this work as invalid PCR results do not produce a numerical value. Further work is required to assess the difference in interference between CG and heparin using quantitative PCR methodology, and to establish a reliable method to reduce carrageenan interference in diagnostic samples.

## *In vivo* carrageenan interference analysis

To further test the carrageenan interference impact in a real-world setting, we explored *in vivo* the effect of carrageenan spray over a 24-hour period (Fig 4). There was a clear increase in Ct values at 15 minutes post-dose, with Ct values returning close to baseline levels at 30 minutes post-dose. A change in IC Ct values of up to 1 is expected due standard variance within a run, so an increase of 3-5 Ct values between baseline and 15 minutes post-dose (when tested on the same run) suggests that there is little CG remaining in the nasal passages after 30 minutes post-dose. Acknowledging the limitations of this experiment being performed on a single individual, we suggest that, as a precaution, any NP swabs intended for diagnostic PCR testing should be collected a minimum of 30 minutes post-dose when administering intranasal CG. Whether prolonged use leads to extension of the time of interference through achievement of a steady state is unknown.

The presence of PCR interference for up to two days after CG dosage as seen in trial partici-pants suggests that the data shown in Fig 4 may be an under-representation of the ability of CG remaining in the nasal passages post-dosage to cause PCR interference. This is not surprising, as the subject of the *in vivo* experiment only administered one dose per experiment, and had not used the product consistently over a period of time, whereas the trial participants indicated that they had been using the product multiple times per day over an extended period.

While more data is required to make a definitive statement, the above examples provide proof-of-concept that CG-mediated PCR interference can occur outside of an experimental laboratory context, and suggests that there may be consequences when diagnostic PCR testing is performed on NP swabs collected from people using CG containing nasal sprays.

## Limitations

The use of laboratory spiked samples creates limitations when assessing PCR interference, as the amount and length of time that CG that remains in the nasal passage after administration is unknown, and the concentrations tested here may be under- or over-representative compared to what would be found in samples from people using intranasal sprays containing CG. However, the examples above do provide confidence that our experimental results are also be applicable in the real world. The *in vivo* pilot study to investigate how long interference persists after administration of CG-based nasal sprays is an n of 1, and must be further investigated before definitive conclusions can be made. In addition, there may be differing effects seen across a population, such as impacts due to underlying health conditions, concomitant medications or other unknown parameters.

## Conclusions

The impact of PCR interference from commercially available CG-containing nasal sprays highlights the importance of a thorough medical history when treating patients to ensure accurate laboratory results are received. In high-risk settings, it would be helpful to adjust NP sampling protocols, for examples collecting swabs a minimum of 30 minutes after using a CG nasal spray, in order to minimize the effect of PCR inhibitors.

We demonstrate here that CG has an inhibitory impact on RT-qPCR, similar to that seen in the presence of heparin (Figs 1–4). To our knowledge, this is the first study to demonstrate CG-mediated PCR interference, and raises a red flag regarding the impact of these products in the context of future public health responses. In addition, it suggests that the results of studies investigating the efficacy of carrageenan-based products using PCR assays should be further scrutinized. Our results also suggest that the use of any nasal products should be consid-ered when looking at the validity of clinical results obtained via PCR test methods with NP samples. However, further clinical studies are required to validate these findings. In addition, more research is required into the mechanistic action of GAGs in PCR interference and spe-cifically, how this interference can be reliably overcome in a diagnostic setting.

## Supporting information

**S1 Table. 95% Confidence intervals (CI) for the CT values of the E-gene from samples presented in Fig 3. Results are presented as the mean of three technical replicates. N/A denotes samples that returned invalid results; CG = carrageenan, Hz = heparinase I.** (PDF)

**S2 Table. 95% Confidence Intervals (CI) of the Ct values from the RdRP/S-gene of samples presented in Fig 3; Results are presented as the mean of three technical replicates. N/A denotes samples that returned invalid results; CG = carrageenan, Hz = heparinase I.** (PDF)

**S3 Table. 95% Confidence Intervals (CI) of the N-gene Ct values from samples presented in Fig 3, Results are presented as a mean of three technical replicates. N/A is present where samples returned invalid results, CG = carrageenan, Hz = heparinase I.**
(PDF)

**S4 Table. 95% Confidence Intervals (CI) for the internal control of samples presented in Fig 3, Results are presented as a mean of three technical replicates. N/A is present where samples returned invalid results, CG = carrageenan Hz = heparinase I.**
(PDF)

**S5 Table. Summary of IC Ct value results from samples which failed IC threshold test multiple times. (i) Initial testing, (r) repeat testing.**
(PDF)

## Acknowledgments

We would like to acknowledge Caitlin McQueen Thomson for proof-reading the article.

## Author contributions

**Conceptualization:** Taylor Corocher, Kira Edwards, Donald Campbell, Paul Monagle, Prahlad Ho.

**Data curation:** Taylor Corocher, Kira Edwards.

**Formal analysis:** Taylor Corocher, Kira Edwards.

**Funding acquisition:** Donald Campbell, Paul Monagle.

**Investigation:** Taylor Corocher, Kira Edwards.

**Methodology:** Taylor Corocher, Kira Edwards.

**Project administration:** Kira Edwards.

**Resources:** Hui Yin Lim.

**Supervision:** Yvonne Hersusianto, Donald Campbell, Hui Yin Lim, Paul Monagle, Prahlad Ho.

**Validation:** Taylor Corocher.

**Visualization:** Taylor Corocher, Kira Edwards.

**Writing – original draft:** Taylor Corocher, Kira Edwards.

**Writing – review & editing:** Taylor Corocher, Kira Edwards, Yvonne Hersusianto, Donald Campbell, Paul Monagle, Prahlad Ho.

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
