## [Decision Letter · Decision Letter 0]

28 Oct 2024

PONE-D-24-31189Over-the-counter Carrageenan-based sprays may interfere with PCR testing of nasopharyngeal swabs to detect SARS-CoV-2PLOS ONE

Dear Dr. Corocher,

Thank you for submitting your manuscript to PLOS ONE. After careful consideration, we feel that it has merit but does not fully meet PLOS ONE’s publication criteria as it currently stands. Therefore, we invite you to submit a revised version of the manuscript that addresses the points raised during the review process.

We look forward to receiving your revised manuscript.

Kind regards,

Shih-Chao Lin, Ph.D.

Academic Editor

PLOS ONE

2. We note that your Data Availability Statement is currently as follows: [All relevant data are within the manuscript and its Supporting Information files]

3. We notice that your supplementary tables are included in the manuscript file. Please remove them and upload them with the file type 'Supporting Information'. Please ensure that each Supporting Information file has a legend listed in the manuscript after the references list.

Reviewers' comments:

Reviewer's Responses to Questions

**Comments to the Author**

1. Is the manuscript technically sound, and do the data support the conclusions?

Reviewer #1: Yes

Reviewer #2: Yes

2. Has the statistical analysis been performed appropriately and rigorously? 

Reviewer #1: No

Reviewer #2: Yes

3. Have the authors made all data underlying the findings in their manuscript fully available?

Reviewer #1: Yes

Reviewer #2: Yes

4. Is the manuscript presented in an intelligible fashion and written in standard English?

Reviewer #1: Yes

Reviewer #2: Yes

5. Review Comments to the Author

Reviewer #1: 1. In line 62, the sentence "resulting in an increase in invalid and false negative result when testing nasopharyngeal (NP) swabs..." should be rewritten to correct grammar issues. Consider changing it to "resulting in an increase in invalid and false-negative results when testing nasopharyngeal (NP) swabs...

2. The introduction mentions the purpose of the study in line 74, but the phrasing is somewhat unclear. It would be better to explicitly state the primary objective in a clearer manner, such as: "This study aims to evaluate the potential PCR interference caused by CG and its implications for public health."

3. (lines 135–143). The description could be expanded to clarify how the nasal spray's potential interference was measured after spiking with SARS-CoV-2.

4. In several places (e.g., lines 165 and 167), the text refers to "Error! Reference source not found." These missing references should be corrected to properly link the figures and tables.

5. In Table 3-1 (lines 185–188), the SARS-CoV-2 positive agreement percentage across different CG brands is presented. However, the significance of the 20% agreement isn’t fully explained. More context is needed regarding what this percentage means and how it affects diagnostic reliability.

6. In the results section (lines 189–200), when comparing CG and heparin interference, there’s a lack of statistical analysis that supports the claim of CG's stronger effect. Adding statistical significance or confidence intervals would make the results more robust.

7. “normalised” (line 214 and 223) should be corrected to "normalized."

8. Sample size of 1 (line 330), which is inadequate for making broader conclusions.

9. Two swab types are used for sample collection during the trial” could benefit from more detail on why this is significant.

10. No mention of controlling for potential confounders in the study participants who used nasal sprays (e.g., other medications, health status).

11. There's no discussion of variability in response to CG among different demographics or populations, which could influence the findings.

Reviewer #2: Mechanistic Insights: While the study demonstrates interference, further exploration into the mechanism by which carrageenan inhibits PCR could enhance understanding and potentially guide mitigation strategies.

Clinical Relevance: It would be beneficial to discuss the clinical implications of these findings more explicitly. How might this affect patient care or testing protocols, especially in high-risk settings?

Comparison with Heparin: The comparison of carrageenan to heparin is interesting but could be elaborated. More detailed data on concentration effects and the specific conditions of the experiments would clarify the relative impact of these substances.

Recommendations:

Wider Implications: Consider including a section discussing potential regulatory implications for OTC products containing carrageenan, especially in the context of respiratory infections.

Future Research Directions: Suggest further studies to explore possible strategies to mitigate the interference of carrageenan in PCR testing.

Clarity and Structure: Ensure that the abstract succinctly summarizes the key findings and their implications, as well as the importance of the research in the context of ongoing public health challenges.

6. PLOS authors have the option to publish the peer review history of their article (what does this mean? ). If published, this will include your full peer review and any attached files.

**Do you want your identity to be public for this peer review?** For information about this choice, including consent withdrawal, please see our Privacy Policy .

Reviewer #1: **Yes: ** Safdar Ali

Department of Microbiology, Cholistan University of Veterinary and Animal Sciences, Bahawalpur, Pakistan

Reviewer #2: No

---

## [Author Response · Author response to Decision Letter 1]

10 Dec 2024

Editor comments

We have reviewed and amended the manuscript to conform with the PLOS ONE style requirements.

2. We note that your Data Availability Statement is currently as follows: [All relevant data are within the manuscript and its Supporting Information files]

We have uploaded the raw data set to Science Data Bank, as noted in lines 40-42.

3. We notice that your supplementary tables are included in the manuscript file. Please remove them and upload them with the file type 'Supporting Information'. Please ensure that each Supporting Information file has a legend listed in the manuscript after the references list.

We have amended the manuscript as requested and uploaded supplementary tables as separate files.

We have reviewed the reference list and none have been retracted. A minor amendment was made to reference [2] to fix a broken link.

Reviewer comments

Reviewer #1:

1. In line 62, the sentence "resulting in an increase in invalid and false negative result when testing nasopharyngeal (NP) swabs..." should be rewritten to correct grammar issues. Consider changing it to "resulting in an increase in invalid and false-negative results when testing nasopharyngeal (NP) swabs...

The sentence has been adjusted as suggested by the reviewer.

2. The introduction mentions the purpose of the study in line 74, but the phrasing is somewhat unclear. It would be better to explicitly state the primary objective in a clearer manner, such as: "This study aims to evaluate the potential PCR interference caused by CG and its implications for public health."

The aim of the paper has been clarified.

3. (lines 135–143). The description could be expanded to clarify how the nasal spray's potential interference was measured after spiking with SARS-CoV-2.

The paragraph was expanded to explain that Ct values of three SARS-CoV-2 specific genes were compared before and after spiking samples with carrageenan.

4. In several places (e.g., lines 165 and 167), the text refers to "Error! Reference source not found." These missing references should be corrected to properly link the figures and tables.

Formatting caused some links between texts and figures to result in an error, which have been corrected.

5. In Table 3-1 (lines 185–188), the SARS-CoV-2 positive agreement percentage across different CG brands is presented. However, the significance of the 20% agreement isn’t fully explained. More context is needed regarding what this percentage means and how it affects diagnostic reliability.

The referencing paragraph has been expanded to include an explanation of why different formulations were tested (due to variation in proportions of carrageenan isotopes across nasal sprays). We have further addressed in the discussion the significance of 20% agreement (lines 394-396), consistent inhibition regardless of the brand of carrageenan suggests the likely hood that any formulation of CG in a nasal spray at similar concentrations will cause interference as demonstrated.

6. In the results section (lines 189–200), when comparing CG and heparin interference, there’s a lack of statistical analysis that supports the claim of CG's stronger effect. Adding statistical significance or confidence intervals would make the results more robust.

We note in lines 310-313 that further work is required to fully understand the degree in difference of interference between carrageenan and heparin. In the result section we note that heparin generally has a higher interference effect compared to CG, as the Ct values are inflated more compared to CG in Fig 3, the mean Ct value in the presence of heparin is higher on average between each gene as demonstrated in S1-3.

Where possible we have included confidence intervals. However, it is difficult to comment on the significance of the CI produced from these results as an invalid result is not numerical. However, in the samples for the 1/8 and 1/32 dilutions the Ct values of the CI is generally higher e.g. at 1/32 dilution the E-gene returns a Ct value of 32.12 in the presence of heparin and 28.01 in the presence of carrageenan, indicating more PCR interference in the presence of heparin.

7. “normalised” (line 214 and 223) should be corrected to "normalized."

The manuscript has been reviewed to ensure it conforms to American English standards.

8. Sample size of 1 (line 330), which is inadequate for making broader conclusions.

We agree with the reviewer, that a sample size of n=1 is inadequate to make further conclusions. The aim of this experiment was to provide in vivo proof of concept that interference does occur. We have reworded the paragraph to clarify.

9. Two swab types are used for sample collection during the trial” could benefit from more detail on why this is significant.

Added a sentence explaining why two swab types were used in line 252-256.

10. No mention of controlling for potential confounders in the study participants who used nasal sprays (e.g., other medications, health status).

Added a sentence at 340-342 that refers to the limitation of other confounders in the study participants.

11. There's no discussion of variability in response to CG among different demographics or populations, which could influence the findings.

Added a line at 340-342 to include the varying demographics as a limitation to the findings.

Reviewer #2:

Mechanistic Insights: While the study demonstrates interference, further exploration into the mechanism by which carrageenan inhibits PCR could enhance understanding and potentially guide mitigation strategies.

In the conclusion a paragraph has been added to address the mechanistic insights of CG as future studies to enable us to reliably overcome GAG PCR interference in a clinical setting.

Clinical Relevance: It would be beneficial to discuss the clinical implications of these findings more explicitly. How might this affect patient care or testing protocols, especially in high-risk settings?

Added a paragraph to address this comment in lines 352-355 of the conclusions.

Comparison with Heparin: The comparison of carrageenan to heparin is interesting but could be elaborated. More detailed data on concentration effects and the specific conditions of the experiments would clarify the relative impact of these substances.

Addressed the comment with changes in lines 311-313 to better clarify the future directions of the work needed.

Recommendations:

Wider Implications: Consider including a section discussing potential regulatory implications for OTC products containing carrageenan, especially in the context of respiratory infections.

A section has been added to the introduction and conclusion on the impact of access to PCR interferents used to prevent/treat respiratory illnesses, and the potential increase in accidental spread of disease as a result.

Future Research Directions: Suggest further studies to explore possible strategies to mitigate the interference of carrageenan in PCR testing.

Addressed the comment with changes in lines 355-358 to better clarify the future directions of the work needed.

Clarity and Structure: Ensure that the abstract succinctly summarizes the key findings and their implications, as well as the importance of the research in the context of ongoing public health challenges.

The abstract has been restructured, results have been summarised more succinctly, and more emphasis has been put on the impact of the work on public health challenges.

---

## [Editor Report · Decision Letter 1]

16 Dec 2024

Over-the-counter Carrageenan-based sprays may interfere with PCR testing of nasopharyngeal swabs to detect SARS-CoV-2

PONE-D-24-31189R1

Dear Dr. Corocher,

We’re pleased to inform you that your manuscript has been judged scientifically suitable for publication and will be formally accepted for publication once it meets all outstanding technical requirements.

Kind regards,

Shih-Chao Lin, Ph.D.

Academic Editor

PLOS ONE

---

## [Editor Report · Acceptance letter]

PONE-D-24-31189R1

PLOS ONE

Dear Dr. Corocher,

I'm pleased to inform you that your manuscript has been deemed suitable for publication in PLOS ONE. Congratulations! Your manuscript is now being handed over to our production team.

Kind regards,

on behalf of

Dr. Shih-Chao Lin

Academic Editor

PLOS ONE